# Quantitative Proteomic Analysis Reveals the Mechanisms of Sinapine Alleviate Macrophage Foaming

**DOI:** 10.3390/molecules28052012

**Published:** 2023-02-21

**Authors:** Aiyang Liu, Bin Liao, Shipeng Yin, Zhan Ye, Mengxue He, Xue Li, Yuanfa Liu, Yongjiang Xu

**Affiliations:** State Key Laboratory of Food Science and Technology, School of Food Science and Technology, National Engineering Research Center for Functional Food, National Engineering Laboratory for Cereal Fermentation Technology, Collaborative Innovation Center of Food Safety and Quality Control, Jiangnan University, 1800 Lihu Road, Wuxi 214122, China

**Keywords:** sinapine, foam cells, proteomics, atherosclerosis, lipid metabolism

## Abstract

Rapeseed polyphenols have cardiovascular protective effects. Sinapine, one main rapeseed polyphenol, possesses antioxidative, anti-inflammatory, and antitumor properties. However, no research has been published about the role of sinapine in alleviating macrophage foaming. This study aimed to reveal the macrophage foaming alleviation mechanism of sinapine by applying quantitative proteomics and bioinformatics analyses. A new approach was developed to retrieve sinapine from rapeseed meals by using hot-alcohol-reflux-assisted sonication combined with anti-solvent precipitation. The sinapine yield of the new approach was significantly higher than in traditional methods. Proteomics was performed to investigate the effects of sinapine on foam cells, and it showed that sinapine can alleviate foam cell formation. Moreover, sinapine suppressed CD36 expression, enhanced the CDC42 expression, and activated the JAK2 and the STAT3 in the foam cells. These findings suggest that the action of sinapine on foam cells inhibits cholesterol uptake, activates cholesterol efflux, and converts macrophages from pro-inflammatory M1 to anti-inflammatory M2. This study confirms the abundance of sinapine in rapeseed oil by-products and elucidates the biochemical mechanisms of sinapine that alleviates macrophage foaming, which may provide new perspectives for reprocessing rapeseed oil by-products.

## 1. Introduction

Rapeseeds are the second-largest source of edible oil production and one of the most important oil crops in the world [1]. Sinapine is the most abundant phenolic compound in rapeseeds, accounting for up to 80% of the total phenolic content [2], and is mainly found in rapeseed meals [3]. Sinapine has some bioactivities, such as anti-inflammatory [4], antioxidant [5], and anti-angiogenic properties [6]. Various polyphenols, such as gallic acid [7] and chlorogenic acid [8], alleviate foam cells and prevent atherosclerosis. Though sinapine is the main polyphenol in rapeseed meals, there are few studies on its alleviation effect on foam cells.

Macrophages are immunological and metabolic cells involved in atherosclerosis occurrence and development [9]. Macrophages take up modified LDL in the form of oxidized LDL particles (ox-LDL) through uptake receptors [10]. Lipid metabolism in macrophages involves three distinct processes: cholesterol uptake, esterification, and efflux. The dysregulation of these lipid processes leads to the formation of lipid-dense macrophages, known as “foam cells”, which are defined as a characteristic of early atherosclerosis [11]. Thus, finding ways to effectively inhibit the formation of foam cells and exploring the mechanism of inhibiting foam cells are significant for the prevention and treatment of atherosclerosis.

In this study, we developed a new extraction method to retrieve sinapine from rapeseed meals by using hot-alcohol-reflux-assisted sonication combined with anti-solvent precipitation. We found a significant ameliorative effect of sinapine on foam cell formation. Through label-free quantitative proteomics (LFQ) and bioinformatics analysis of differential proteins and the pathways involved, we found that sinapine can activate the CD36/CDC42-JAK2-STAT3 pathway in foam cells. We achieved the efficient utilization of rapeseed-oil by-products and revealed the main plant polyphenols in rapeseed meals, sinapine, to alleviate macrophage foaming.

## 2. Results

### 2.1. The Effect of Extraction Methods on Sinapine

The total content of sinapine was measured according to the equation of the standard curve: y = 0.0515x − 0.0047. The sinapine contents in rapeseed meals prepared from HE, HE + UE, and HE + UE + AP were analyzed. The sinapine yields of the three methods were 6.69 ± 0.072, 10.92 ± 0.007, and 15.76 ± 0.015 mg/g, respectively. The sinapine content in rapeseed meals prepared from HE + UE + AP was significantly higher than those of the other methods (Figure 1).

### 2.2. Regulatory Effects of Sinapine on Cholesterol Accumulation in Foam Cells

Foam cells were treated with different concentrations of sinapine (0–640 μM) for 24 or 48 h. After that, cell viability was detected with a CCK8 kit. The concentration range of sinapine that did not significantly affect the cell viability of macrophages was determined. In the range of 0–80 μM, the cell viability was not significantly affected when macrophages were treated with sinapine for 24 h (Figure 2A). Therefore, the effect of sinapine on macrophages was further investigated at a concentration of 80 μM after 24 h of incubation. The Oil Red O staining results demonstrated the intracellular lipid content greatly decreased after adding sinapine (80 μM) for 24 h (Figure 2B). The semi-quantitative Oil Red O assay showed that treatment with sinapine significantly reduced lipid droplet content in foam cells (Figure 2C).

After a treatment under optimal conditions, the amounts of TC, FC, and CE/TC were measured in foam cells. In the sinapine group, the cells contained significantly more FC than the ox-LDL group did (3.607 ± 0.041 vs. 1.037 ± 0.174 mg/g). The TC and CE/TC were lower in the AST group than in the ox-LDL group (5.187 ± 0.256 vs. 11.163 ± 0.184 mg/g, and 29.795 ± 3.523% vs. 85.2 ± 1.671%, respectively) (*p* < 0.05) (Figure 2D).

### 2.3. Differences in Differentially Expressed Proteins (DEPs) between Different Treatments

In total, 3554 proteins were found to be differentially expressed among the control, the ox-LDL, and the sinapine groups by screening DEPs in foam cells with LFQ-based quantitative proteomics. A principal component analysis (PCA) was used to examine the protein expression changes induced by ox-LDL and sinapine interventions. The protein profiles of the control, the ox-LDL, and the sinapine groups differed significantly. The protein expressions were significantly different between the ox-LDL group and the control group. In addition, the sinapine intervention altered the protein expression profile compared to the ox-LDL group (Figure 3A,B). To further compare the control and ox-LDL groups with the sinapine group, we screened out the significantly different proteins by using the following criteria: fold change (FC) > 1.2, *p* < 0.05 for upregulated proteins; FC < 0.83, *p* < 0.05 for downregulated proteins. In comparing the ox-LDL group and the control group, a total of 2390 significant DEPs and 512 DEPs proteins were identified in the sinapine group compared to the ox-LDL group. Thus, sinapine treatment significantly affected protein expression in foam cells. Specifically, as seen in the volcano plot, when comparing the ox-LDL group to the control group, 2368 DEPs were downregulated significantly, and 20 DEPs were significantly upregulated, whereas 271 DEPs were downregulated significantly. Sinapine induced the upregulation of 214 DEPs compared with the ox-LDL group (Figure 3C). As a result, the sinapine group had more upregulated proteins than downregulated proteins. These data indicate that the main biological effects of sinapine are the upregulation of proteins and the downregulation of foam cells. The above results suggest that protein expression in foam cells affected by ox-LDL is compensated by sinapine treatment and that sinapine significantly alleviates macrophage foaming.

### 2.4. DEPs Enriched in Gene Ontology (GO) and Kyoto Encyclopedia of Genes and Genomes (KEGG)

In total, 2390 DEPs in the control group versus the ox-LDL group, and 512 DEPs in the ox-LDL group versus the sinapine group were identified (Figure 4A). Among them, 344 proteins were common in the control group, the ox-LDL group, and the AST group (Figure 4A). A bioinformatics analysis of these DEPs was performed to find the proteins that play important roles in the sinapine-treating process. GO and KEGG databases were accessed for the 344 DEPs among the control, the ox-LDL, and the sinapine groups. The MF, CC, and BP are shown according to GO annotations (Figure 4B). The DEPs of MF included binding, catalytic, structural molecule, molecular function regulator, transporter, ATP-dependent, transcription regulator, molecular transducer, translation regulator, cytoskeletal motor, and low-density lipoprotein particle receptor activities. In the CC class, the cellular, anatomical entity, protein-containing complex, and BP categorization activities revealed that these DEPs were involved primarily in the cellular process, metabolic process, biological regulation, localization, response to stimulus, multicellular organismal process, immune system process, the biological process involved in interspecies interaction between organisms, developmental process, locomotion, biological adhesion, biological phase, and the reproductive process.

Based on the shared DEPs among the control, ox-LDL, and sinapine groups, an enrichment analysis of the 9 KEGG pathway and foam cell cholesterol metabolism pathway was conducted (Figure 4C).

### 2.5. Effects of Sinapine on the Cholesterol Metabolism in Foam Cells

Whether sinapine regulated cholesterol metabolism in foam cells was investigated. The KEGG pathway analysis revealed that some proteins were involved in those pathways, and the STRING analysis provided evidence of interactions in these proteins. Those proteins interacted and were closely linked, and the expression trends of these proteins in the sinapine group were opposite to those in the ox-LDL group (Figure 5A,B). Among them, CD36, CDC42, JAK2, and STAT3 proteins were closely associated with atherosclerosis and foam cells, and their protein expressions were opposite to those of the ox-LDL group and converged with those of the control group after the sinapine base intervention (Figure 5C,D). Those proteins are key proteins in cholesterol metabolism and foam cell cholesterol efflux, respectively.

### 2.6. Western Blot Validation of Key Proteins

A Western blot of key proteins was conducted to validate the LFQ-based quantitative proteomics. CD36, CDC42, JAK2, and STAT3 DEPs participated in the KEGG pathways and were all involved in the cholesterol metabolism pathway. Consequently, proteins related to sinapine that prevented foam cell formation were verified by Western blot. Western blot (Figure 6A) showed that the relative expression level of CD36 in the ox-LDL group was higher than in the control group. The relative expression levels of CDC42, JAK2, and STAT3 were lower than in the control group, and the relative expression level of CD36 in the sinapine group was lower than in the ox-LDL group. The relative expression levels of CDC42, JAK2, and STAT3 were higher than in the ox-LDL group. These results were confirmed by proteomic analysis. The expressions of those proteins were strikingly higher in the sinapine group than in the ox-LDL group, and this is consistent with the proteomic analysis.

## 3. Discussion

Although sinapine can significantly improve several chronic diseases, limited research has focused on its beneficial effects on the prevention and intervention of atherosclerosis. Experiments here first suggest that sinapine can prevent and intervene with atherosclerosis by modulating lipid accumulation in foam cells induced by ox-LDL. In addition, CD36, JAK2, STAT3, and CDC42 are essential in reducing lipid accumulation and atherosclerosis in foam cells.

In the arteries of early atherosclerotic lesions, macrophages accumulate under the endothelium [12]. Moreover, macrophages are rich in lipids due to the differentiation of monocytes into macrophages, which are taken up by modified lipoproteins [10]. Lipid metabolism in macrophages involves three distinct processes: absorption, esterification, and excretion of cholesterol. These dysregulated lipid pathways lead to foam cell formation, which results in various atherosclerotic effects, including matrix degradation. When located in this stage, macrophages lead to plaque rupture [13]. Research with animal models shows that foam cell death can lead to atherosclerosis. Moreover, atherosclerosis occurs in the cerebral arteries, coronary arteries, and aorta of humans [14]. Thus, the formation of foam cells is one of the targets for the treatment of atherosclerosis. According to the lipid and inflammation hypothesis, anti-inflammatory therapies and lipid-modulating strategies remain the backbone of atherosclerosis treatment.

Several types of cells express CD36, including monocytes and macrophages. CD36 has been implicated in atherosclerosis by promoting foam cell formation in the intima of blood vessels [15,16,17]. CD36 has several functions in regulating modified LDL binding, inflammatory processes, lipid metabolism, fatty acid transport, and immunity [15]. In atherosclerosis, ox-LDL can be taken up by macrophages through CD36 and transformed into foam cells, secreting inflammatory cytokines and chemokines [18]. Specifically, a decrease in CD36 protein expression reduces the uptake of ox-LDL and further inhibits the formation of foam cells and the development of atherosclerotic plaques in ApoE^-/-^mice [19]. The expression of CD36 in macrophages is upregulated by several proatherogenic stimuli, such as ox-LDL. Moreover, ox-LDL enhances the interaction of CD36 with JAK2, induces phosphorylation of JAK2, and subsequently activates STAT3 signaling [20]. The Janus kinase/signal transducer and activator of transcription (JAK/STAT) is involved in regulating mammalian cell proliferation and differentiation and various physiological functions [21]. Receptor binding triggers autophosphorylation to activate the Janus kinase (JAK), which activates the STAT3 transcription factor (signal transducer and activator of transcription), translocates to the nucleus and binds DNA to regulate transcription [22]. The JAK2/STAT3 signaling pathway can regulate lipid metabolism disorders caused by environmental pollutants [23]. The JAK2/STAT3 signaling pathway is also closely related to lipid metabolism. Mice deficient in JAK2 have lipolysis and impaired insulin resistance [24]. Moreover, activating the JAK2/STAT3 pathway in macrophages can reduce macrophage lipid levels [25], suggesting that the JAK2/STAT3 pathway may play a role in reducing intracellular lipid accumulation. Therefore, the addition of sinapine reduces the expressions of CD36, JAK2, and STAT3, thereby reducing the ox-LDL uptake and lipid accumulation in macrophages.

Additionally, the protein expression of the cell-division cycle 42 (CDC42) in foam cells treated with sinapine is significantly increased. CDC42 affects cytoskeletal rearrangement and lipid metabolism in macrophages. It promotes actin polymerization and disrupts lipid rafts [26,27]. The CDC42 protein directly interacts with ABCA1 to direct lipid transport out of cells [28]. The small GTPase CDC42 is central to the rearrangement of cytoskeletons and lipid metabolism [29]. AIBP activates CDC42 expression to increase cholesterol efflux [30], and CDC42 may be involved in cholesterol vesicle transport across the trans-Golgi network and plasma membrane, suggesting that CDC42 activation induced by APOA-I enhances vesicular transport, prompting ABCA1 to export cholesterol from cells [31]. The activation of CDC42 causes lipid raft abundance and structure changes and the rearrangement of actin filaments during actin remodeling, resulting in raft clustering. Lipid raft changes create regions of the plasma membrane that can bind apoA-I. The increased apoA-I binding leads to the intracellular redistribution of cholesterol to the plasma membrane and, together with increased access to cholesterol, can enhance cholesterol efflux [30]. Therefore, after the foam cells were treated with sinapine, the CDC42 protein expression significantly increased, indicating that sinapine can promote the efflux of cholesterol in foam cells.

Interestingly, CDC42 represents a critical source of signals that promote STAT3 activation [32]. Human cells expressing mutationally activated CDC42 can activate STAT3 due to phosphorylation at tyr705 and ser727 [33]. Inhibition experiments indicate that CDC42 activate STAT3 though JAK2 [34]. Activation of CDC42 affects the protein expressions of JAK2 and STAT3, which is consistent with our study. Macrophages found in atherosclerotic plaques can alter their phenotypes in response to various stimuli. Modified LDL and cholesterol crystals stimulate pro-inflammatory M1 macrophages [35]. M1 macrophages produce and secrete pro-inflammatory cytokines, such as TNF-α, IL-1β, IL-6, NO, and ROS [36]. M1 macrophages also express different chemokine receptor ligands, such as CXC chemokine ligand CXCL-9, CXCL-10, and CXCL-5. CXCL-5 promotes the recruitment of Th1 and natural killer cells, which are important in killing intracellular pathogens [37]. Thus, M1 macrophages have potent antimicrobial effects. However, in the aseptic inflammatory setting of AS, pro-inflammatory M1 macrophages lead to a sustained inflammatory response, resulting in peripheral tissue damage [38]. M2 macrophages are activated by IL-4 and IL-13 and secrete IL-10, TNF-α, CCL-17, and CCL-22. Therefore, the main roles of M2 macrophages are to prevent tissue damage, exert anti-inflammatory effects and promote tissue repair [39]. CD36 is the main scavenger receptor for ox-LDL phagocytosis by macrophages [40], suggesting that M1 macrophages can phagocytose large amounts of ox-LDL and facilitate lipid accumulation. This finding is consistent with the present study, whereas M2 macrophages lowly expressed LXRα, ABCA1, and ApoE B. The above results suggest that M1 macrophages are more prone to foam cell formation than M2 macrophages are [35]. In addition to anti-hemorrhagic properties, M2 macrophages can prevent foam cell formation. Reportedly, the activation of JAK/STAT1 pathway may lead to the polarization of pro-inflammatory macrophages into M2 [41]. Moreover, the JAK/STAT3 pathway can shift the macrophage phenotype from M1 to M2, inhibiting atherosclerotic lesions in early and late development stages [42,43]. Therefore, sinapine induces the conversion from M1 to M2 macrophages, thereby inhibiting foam cell formation.

## 4. Materials and Methods

### 4.1. Materials

Sinapine was purchased from Solarbio (purity ≥ 96%, Beijing Solar Science Technology Co., Ltd., Beijing, China). Ox-LDL was bought from Yiyuan (Yiyuan Co. Ltd., Guangzhou, China). Antibodies were offered by Proteintech (Proteintech Group, Inc., Wuhan, China). The bicinchoninic acid (BCA) protein quantification kit was purchased from Beyotime (Beyotime Institute of Biotechnology, Shanghai, China).

### 4.2. Preparation of Rapeseed-Meal Samples and Extraction of Sinapine

Rapeseed-meal samples were prepared by removing any visible debris, crushed, passed through a 60-mesh sieve, and dried in an oven at 40 °C to a constant weight. The rapeseed meal produced in the third step of the process was divided into three equal parts with a weight of 1.25 g, compacted in a filter paper sleeve, and immersed in a round-bottom flask with a Soxhlet extractor (Soxtec-2055, FOSS, Eden Prairie, MN, USA) containing petroleum ether. Then the three parts were defatted at reflux in a water bath at 70 °C to a constant weight and processed in three separate ways. (A) Hot ethanol extraction (HE): The oiled rapeseed meal was extracted four times (one hour each time), at 90 °C, with 130 mL of 80% ethanol, under hot reflux, and the resulting ethanol extract was poured into a round-bottom flask and concentrated to 10 mL under vacuum to obtain the final extract. (B) HE with ultrasonic extraction (HE + UE): The de-oiled rapeseed meal was extracted under hot reflux, at 90 °C, with 130 mL of 80% ethanol four times (1 h each time) and ultrasonicated with an ultrasonic power of 600 W during the last 40 min. The resulting ethanol extract was poured into a round-bottom flask and concentrated to 10 mL under vacuum to obtain the final extract. (C) HE + UE combined with anti-solvent precipitation (HE + UE + AP): The de-oiled rapeseed meal was extracted under hot reflux, at 90 °C, with 130 mL of 80% ethanol four times (1 h each time) and ultrasonicated with an ultrasonic power of 600 W during the last 40 min. The resulting ethanol extract (petroleum ether) was poured into a round bottom flask and concentrated to 10 mL under vacuum to obtain the final extract. The mustard bases were precipitated with the anti-solvent method. Next, 50 mL of a counter solvent was put in a beaker and placed on a magnetic stirrer at the speed of 700–800 rpm. Then 10 mL of the extraction solution was taken at a counter solvent to an extraction solution ratio of 5:1 and dropped into the beaker at a rate of 4 mL/min to obtain a mixed suspension, which was then centrifuged at 4000 rpm for 10 min. The sample was prepared after the supernatant was removed. Standard sinapine was used to authenticate the UV-visible spectrum (UV-1801, Rayleigh, Beijing, China) absorption spectra and calibration curves. Sinapine was detected at 326 nm, and the sinapine extraction yield was calculated. Experiments were repeated three times. The sinapine yield (mg/grapeseed meal) was calculated from Equation (1):(1)$$\mathrm{Yield}=\frac{C_{sinapine}\times V_{solvent}}{m_{\mathrm{rapeseed}\;\mathrm{meal}}}\;$$
where *m*_rapeseed meal_ is the mass of dry matter in rapeseed meal (g).

### 4.3. Foam Cell Formation of Macrophages

THP-1 cells were purchased from Procell Life Technology Co., Ltd. (Wuhan, China)). Under standard culture conditions (5% CO_2_ and 37 °C), the cells were cultured in an RPMI-1640 medium that contained 0.1% non-essential amino acids, 10% fetal bovine serum, and 1% penicillin/streptomycin. Before the experiment, the THP-1 cells were treated with phorbol-12-millistate-13-acetate (Beyotime Institute of Biotechnology, Shanghai, China) for 48 h to differentiate into macrophages.

The THP-1 macrophages were treated with 50 g mL^−1^ ox-LDL and cultured in a 1640 medium containing 5% fetal bovine serum for 48 h to establish a macrophage foam model.

### 4.4. Confirmation of Sinapine Effects on Macrophages

Sinapine was added to foam cells at 20, 40, 80, 160, 320, and 640 μM. After successfully incubating the induced foam cells at 37 °C for 24 or 48 h, cell viability was measured using a CCK8 kit. Afterward, to study the effects of sinapine on the lipid content of macrophages, sinapine (80 μM) was added and cultured with foam cells at 37 °C for 24 h. Cells were harvested after culture, fixed with paraformaldehyde, and stained with Oil Red O-kit. Then Oil Red O staining was measured semi-quantitatively by extracting Oil Red O stain with isopropanol (100%, Alading Biotechnology Co., Ltd., Shanghai, China) for 10 min. After gentle rocking, the absorbance at 492 nm was read [44]. After the treatment with the optimal conditions, the TC, FC, and the ratio of cholesterol ester and total cholesterol (CE/TC) in the cells were determined; the CE was calculated by subtracting FC from TC.

### 4.5. Preparation Method for Proteomic Samples

Three groups of cell samples, namely the control, ox-LDL, and AST groups, were used. Cell pellets were washed three times with a Hanks solution (Thermo Fisher Scientific, Waltham, MA, USA), and cells were scraped and stored at −80 °C for proteomic analysis. Each set of samples was repeated five times.

### 4.6. Protein Identification, Quantitation, and Bioinformatic Analysis

Cells were lysed for 10 min in an ice bath, using a radio immunoprecipitation assay (RIPA) lysis buffer containing 1 mM phenylmethanesulfonyl fluoride (PMSF) (Beyotime Institute of Biotechnology, Shanghai, China) protease inhibitor. After centrifugation (20,000× *g*) at 4 °C for 30 min, a BCA protein quantification kit was used to measure the protein levels in the supernatant.

A sample of protein (about 200 μg) was collected. The volume of precooled acetonitrile(ACN) (MREDA, Beijing, China) was added five times, and the sample was stored at −20 °C for 3 h. The sample was centrifuged to extract the precipitate, and 100 μL of guanidine hydrochloride (Sangon Biotech, Shanghai, China) was added. Then the proteins were reduced with dithiothreitol (DTT) (20 mM, Alading Biotechnology Co., Ltd., Shanghai, China) for 30 min at 56 °C and added with iodoacetamide (IAA) (50 mM, Alading Biotechnology Co., Ltd., Shanghai, China). The reaction proceeded in the dark for 30 min. The sample was transferred to a 10 kDa ultrafiltration tube where 50 μL NH_4_HCO_3_ (50 mM, Alading Biotechnology Co., Ltd., Shanghai, China) and trypsin were added at an enzyme: protein ratio of 1:50. The sample was left overnight to react at 37 °C. After 24 h, the sample was centrifuged at 14,000× *g*, and the filtrate was collected, washed once with a 0.1% aqueous solution, and centrifuged again. Next, 50 μg of each peptide sample was collected, combined into a mixed sample, fractionated using HPLC (AB SCIEX, Foster, CA, USA) (pH = 10.0), and divided into 12 fractions. IDA acquisition was used to identify the fractionated samples for library building and SWATH acquisition mode for data collection of samples. The peptides were resuspended in 0.1% FA and separated using an Acclaim PepMap C18 analytical column in an Ekisgent. The peptide was eluted using a gradient solution comprised of solvent A (98% H_2_O/0.1% FA) and solvent B (98% ACN/0.1% FA). Solutions of 5%, 6%, 27%, 50%, 80%, 80%, 5%, and 5% of solvent B were combined with corresponding volumes of solvent A and applied at 0, 1, 50, 65, 65.5, 75, 75.5, and 90 min. The flow rate was 5 μL/min. A Triple TOF 6600+ (AB SCIEX, Danaher, Washington, DC, USA) was set to 320 °C, and 2.2 kV DDA (data-dependent mode) was selected to switch automatically between MS and MS/MS to collect the spectrum. The MS1 scan ranged from 300 to 1300 *m*/*z*. The scanning range of MS/MS was from 100 to 1500 *m*/*z*.

The graded peptide samples were subjected to IDA acquisition, using a 90 min gradient, and the obtained data were merged on ProteinPilot (http://www.absciex.com/products/software/proteinpilot-software, accessed on 6 January 2023) for a library search to construct the SWATH database. The mixed samples were subjected to IDA data acquisition, and dynamic windows were constructed according to the ion distribution density. The SWATH acquisition method was established for each sample according to the constructed dynamic windows. The results of the ProteinPilot search were used as a library for the analysis of SWATH data, and the relative quantitative information of all the identified proteins was obtained in total.

### 4.7. Western Blotting

Western blotting was performed using the method of Nie [45]. Extracted protein samples (1 mg·mL^−1^), loading buffer, and reducing agent (Thermo Fisher Scientific Inc., Waltham, MA, USA) were mixed and heated at 70 °C for 10 min to denature the protein. The protein samples (20 μL) were separated electrophoretically (NuPAGE 4–12% Bis-Tis GEL; Thermo Fisher Scientific, Waltham, MA, USA) and electrotransferred onto polyvinylidene difluoride membranes (IBLOT2, Thermo Fisher Scientific, Waltham, MA, USA). The membranes were blocked with TBST containing bovine serum albumin (5%, Calbiochem, Billerica, MA, USA) for 2 h at room temperature. The samples were incubated with a primary antibody solution of anti-β-actin (1:1000, Abcam, Cambridge, MA, USA), anti-CD36 (1:1000, Abcam), anti-CDC42 (1:1000, Abcam), anti-JAK2 (1:1000, Abcam), and anti-STAT3 (1:1000, Abcam) overnight at 4 °C. The second antibody (1:1000, Abcam) solution was kept at room temperature for 2 h in a dark room. All the antibodies were diluted with TBST-0.5% BSA dilute solution. β-actin was used as an internal reference protein. An ECL working solution was then applied for 2 min to each blot. ImageJ was used to analyze the protein content’s luminous intensity (Bethesda Softworks LLT, Bethesda, MD, USA).

### 4.8. Data Analysis

Gene Ontology enrichment analysis was used to categorize DEPs into BP, MF, or CC. Metabolic pathways were identified by KEGG (KEGG, http://www.genome.jp/kegg/, accessed on 6 January 2023), Graphpad Prism (Graphpad Prism 8.3.0, Graphpad Software, Boston, MA, USA), or TB tools (v1.068, http://www.tbtools.com/, accessed on 6 January 2023), and MetaboAnalyst (https://www.metaboanalyst.com, accessed on 6 January 2023) visualized changes in proteins after different treatments. The PPI network of atherosclerosis-related DEPs was constructed using the String database (https://string-db.org/, accessed on 6 January 2023), with DEPs as the node and interactions as the edge. For routine data analysis, one-way analysis of variance (ANOVA) (*p* < 0.05) was used to determine statistical significance between processes on SPSS, and we averaged data from five measurements. We also present the data as mean ± standard deviation. For DEPs, it was concluded that there was a fold change > 1.2 or < 0.83 at a *p* < 0.05.

## 5. Conclusions

The ultrasonic hot alcohol reflux method combined with the anti-solvent method was used to extract sinapine from rapeseed for the first time. The sinapine yield of this method was significantly higher than that of the traditional hot alcohol reflux method, and the high-efficiency utilization of rapeseed by-products was realized. Our proteomic findings showed that sinapine reduced foam cell formation in human macrophages by reducing the expression of the scavenger receptor CD36. In addition, sinapine could increase the protein expression of CDC42, which promoted cholesterol efflux in foam cells. In addition, sinapine could also block the formation of foam cells by activating the JAK/STAT3 pathway, converting pro-inflammatory M1 to anti-inflammatory M2 macrophages. Therefore, sinapine is effective at alleviating macrophage foaming and may play a significant role in preventing heart disease in the future.

## Figures and Tables

**Figure 1 molecules-28-02012-f001:**
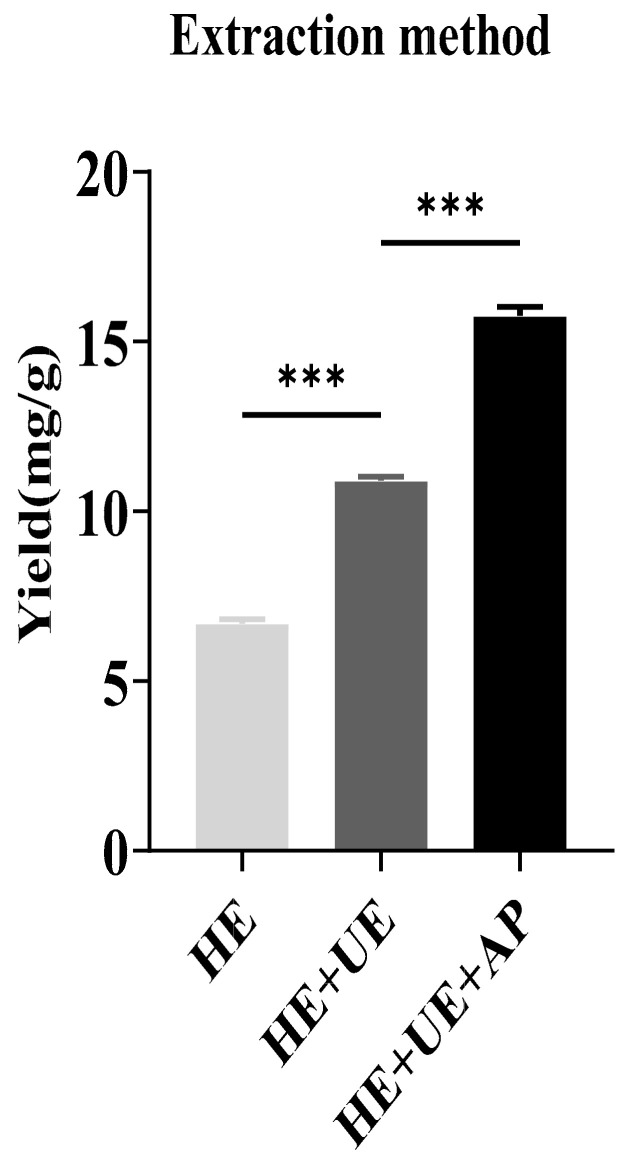
Analysis of sinapine yields in rapeseed meal prepared from different extraction techniques. (HE, hot-ethanol extraction; HE + UE, hot-ethanol extraction and ultrasonic extraction; HE + UE + AP, hot-ethanol extraction with ultrasonic extraction combined with anti-solvent precipitation). Note: *** *p* < 0.001.

**Figure 2 molecules-28-02012-f002:**
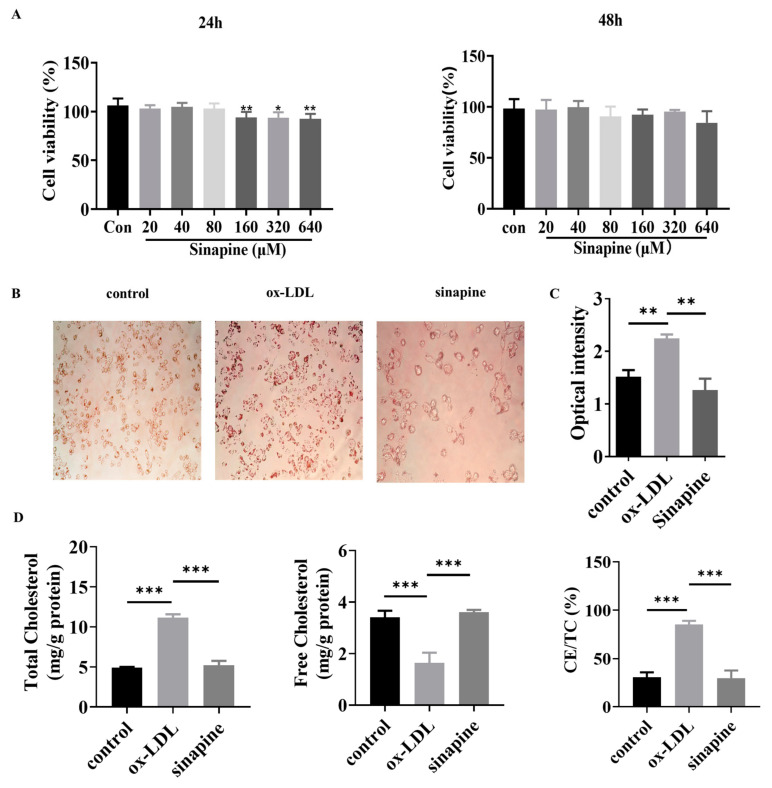
Effects of sinapine on foam cells. (**A**) Effect of sinapine concentration (0–640 μM) on cell viability. (**B**,**C**) Oil Red O staining results and semi-quantitative analysis. (**D**) Effect of sinapine (80 μM) on total cholesterol (TC), free cholesterol (FC), and ratio of cholesterol ester to total cholesterol (CE/TC) in macrophages. Note: * *p* < 0.05, ** *p* < 0.01, and *** *p* < 0.001.

**Figure 3 molecules-28-02012-f003:**
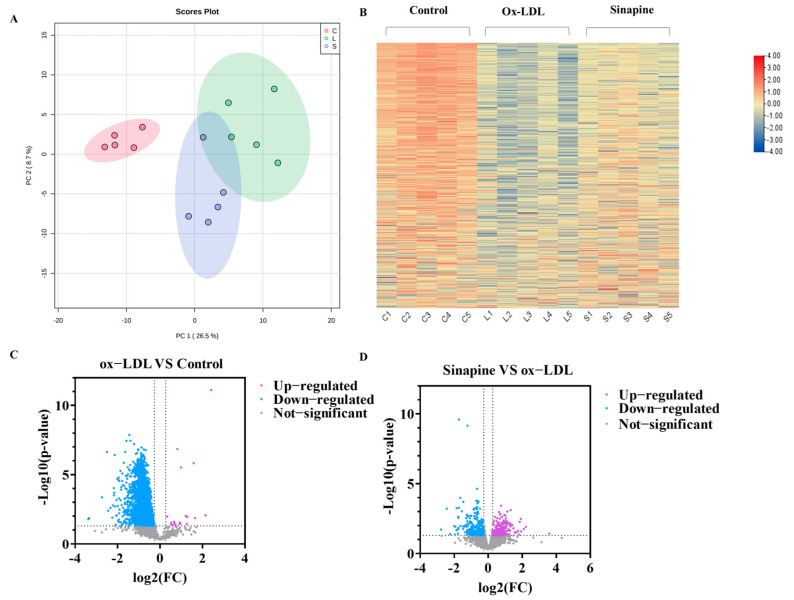
Comparison of DEPs among treatments. (**A**) PCA plots of proteins in foam cells (C, control group; L, ox−LDL group; A, sinapine group). (**B**) The abscissa in the DEP heatmap represents different groups (C1, C2, C3, C4, and C5 = control group; L1, L2, L3, L4, and L5 = ox−LDL group; S1, S2, S3, S4, and S5 = sinapine group), color blocks at different positions indicate the relative levels of protein expression, with red indicating high expression and blue indicating low expression. (**C**) Volcano plots of ox−LDL group vs. control group. (**D**) Volcano plots of sinapine group vs. ox−LDL group.

**Figure 4 molecules-28-02012-f004:**
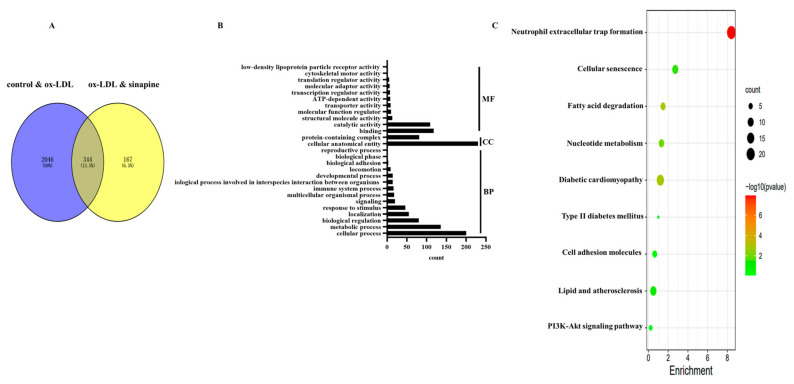
Bioinformatics analysis of DEPs in the control group, ox-LDL group, and sinapine group. (**A**) Venn plot of identified control vs. ox-LDL and ox-LDL vs. sinapine proteins in foam cells. (**B**) GO enrichment analysis of DEPs. (**C**) The above pathways were significantly enriched according to KEGG pathway enrichment analysis (*p* < 0.05). Gradual changes from green to red indicate progressively lower *p*-values, colors indicate significant enrichment levels, and the size of the circles indicates the number of proteins.

**Figure 5 molecules-28-02012-f005:**
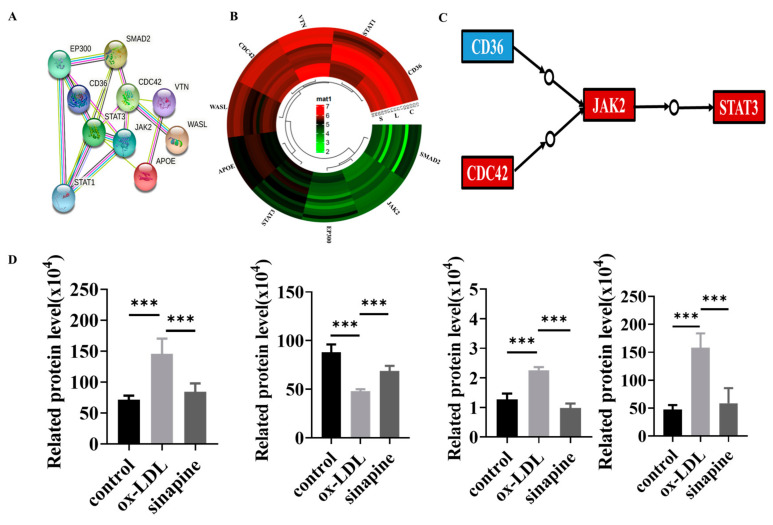
Protein changes in the affected KEGG pathway after sinapine treatment of foam cells. (**A**) Protein–Protein Interaction (PPI) network analysis, with proteins represented by nodes. (**B**) Heatmap of DEPs related to lipid metabolism pathway. (**C**,**D**) Regulatory pathways of key proteins and changes in protein expression in proteomics. Red and blue indicate significantly upregulated and downregulated DEPs, respectively. Note: *** *p* < 0.001.

**Figure 6 molecules-28-02012-f006:**
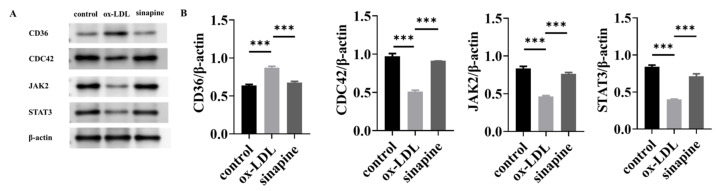
Expression and validation of key proteins after sinapine treatment in foam cells. (**A**) Effects of sinapine on protein levels of CD36, CDC42, JAK2, and STAT3 determined by Western blotting. (**B**) Relative protein content of CD36, CDC42, JAK2, and STAT3 protein expressions was calculated by grayscale analysis with ImageJ. Note: *** *p* < 0.001.

## Data Availability

All the data generated during this study are included in this article.

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
