# Peer review of "Quantitative Proteomic Analysis Reveals the Mechanisms of Sinapine Alleviate Macrophage Foaming"

_molecules, 2023, doi:10.3390/molecules28052012_

Round 1

Reviewer 1 Report

While I consider this work valid, the manuscript has not been carefully prepared and requires considerable editing. The most important critical issues are summarized below.

-The language is poor in some parts of the text and must be reviewed. A few examples:

-lines 15-18 and 58-64. These two sentences can't be read if the authors do not put a dot somewhere. 

line 62: "based the results" should read "based on these results"

-line 74 "higher than that of...."

-lines 200-203: must be deleted.

-line 370: this sentence is not complete

-lines 387-401: must be deleted.

and so on.....

Figure 1: here they indicate Abs at 326 nm while in the text is 330 nm. This small difference in wv, obviously does not affect the data but it is a matter of consistency. Moreover, I must honestly underline that it never happened in my lab that the linearity is still preserved in the Abs range 1,5-2,0. Thus, I have some doubts about this calibration curve with  an r2 of 0,999.

- Par. 2.1. The authors state that the yield of sinapine with the three methods was in the range 0,66- 1,57 %. I do not understand how is this percent expressed. In other words, did they ever use a "control" ? I mean, did they ever submit a known amount of standard sinapine to the same procedure ?Why express the content of sinapine as a % rather than a concentration ? The authors must clarify these points.

-line 101. "analysis......" This sentence means nothing. Please rewrite.

-lines 370-375: Information about antibodies (and the dilution used) is totally missed. The authors must add these data.

-It is not clear to me if the data relative to the extraction of sinapine are relative to a single experiment or not. It seems to understand that only proteomic experiments have been repeated. The authors must clarify this point.

Author Response

Dear Editors and Reviewers,

Thank you for your letter and for the reviewers’ comments concerning our manuscript entitled “Quantitative proteomic analysis reveals the mechanisms of sinapine alleviate macrophage foaming”. Those comments are very helpful for revising and improving our paper. We have studied all issues mentioned in the comments carefully and have corrected. Besides, we have had a check on the whole paper and tried our best to make it better. We have revised the format of the references as required. We hope these meet with approval. Revised portions are marked in red in the manuscript. The main corrections in the paper and the responses to the reviewer’s comments are attached to this revised submission. In addition, all the modifications are highlighted in red in the attachment “main-highlighted”

Below is our careful point-by-point response.

Yours sincerely,

Aiyang Liu

Response to Reviewer 1

Comments

Point 1: The language is poor in some parts of the text and must be reviewed. A few examples:

-lines 15-18 and 58-64. These two sentences can't be read if the authors do not put a dot somewhere.

Response 1: Thank you very much for your comments. We have modified the problem as requested. All the changes and improvements are marked in red color in the revised manuscript. (Line16-19, Line 56-63)

Point 2: line 62: "based the results" should read "based on these results"

Response 2: Thank you very much for your comment. We have now modified this into: “based on these results”. The modifications are highlighted in red in the pdf file “main-highlighted”.(Line 59)

Point 3: line 74 "higher than that of...."

Response 3: Thank you very much for your comment. We have now modified this into: “higher than that of”. The modifications are highlighted in red in the pdf file “main-highlighted”.(Line 73-74)

Point 4: lines 200-203: must be deleted.

Response 4: Thank you very much for your comment. We apologize for our indiscretion. The part has been deleted.

Point 5: line 370: this sentence is not complete

Response 5: Thank you very much for your comments. We have carefully revised the manuscript according to your comments. The changes and improvements are marked in red color in the revised manuscript. (Line 375)

Point 6: lines 387-401: must be deleted.

Response 6: Thank you very much for your comment. We apologize for our indiscretion. The part has been deleted.

Point 7: Figure 1: here they indicate Abs at 326 nm while in the text is 330 nm. This small difference in wv, obviously does not affect the data but it is a matter of consistency.

Response 7: Thank you very much for your comment. We have finally determined the Abs is 326 nm. We apologize for our indiscretion, and we have modified this.

Point 8: Moreover, I must honestly underline that it never happened in my lab that the linearity is still preserved in the Abs range 1,5-2,0. Thus, I have some doubts about this calibration curve with an r2 of 0,999.

Response 8: Thank you very much for your comment. The raw data of the standard curve is plotted below. And perhaps because experimental, instrument condition and other reasons, causes our experimental results to differ from those of other laboratories, but this result is based on several experiments, and it is our original results.

Concentration of sinapine (mg/mL)

absorbancy

1.77

0.091

5.31

0.252

8.85

0.442

12.39

0.649

15.93

0.836

19.47

0.996

23.01

1.195

26.55

1.35

30.09

1.513

33.63

1.711

37.17

1.938

Point 9: Par. 2.1. The authors state that the yield of sinapine with the three methods was in the range of 0,66- 1,57 %. I do not understand how is this percent expressed. In other words, did they ever use a "control"? I mean, did they ever submit a known amount of standard sinapine to the same procedure? Why express the content of sinapine as a % rather than a concentration? The authors must clarify these points.

Response 9: Thank you very much for your comment. The sinapine yield (mg/grapeseed meal) is a key parameter for finding the adequate operating conditions of the extraction process [1]. The formula for sinapine yield is as follows:

Yield=C sinapine * V solvent/m rapeseed meal

And we multiply the yield result from the above formula by 100%, but to be consistent with the other studied’s units, we have changed the units of yield from (%) to (mg/g). We have added this formula to the manuscript, the modifications are highlighted in red in the pdf file “main-highlighted”. (Line 314-316)

    And we have compared the yield of sinapine base in the three extraction methods and based on those methods we have found the optimal extraction method, so there is no need to set the control.

Point 10: line 101. "analysis......" This sentence means nothing. Please rewrite.

Response 10: Thank you very much for your comment. We have modified the problem as requested. All the changes and improvements are marked in red color in the revised manuscript. (Line 107-109)

Point 11: lines 370-375: Information about antibodies (and the dilution used) is totally missed. The authors must add these data.

Response 11: Thank you very much for your comment. We have added the information about antibodies (and the dilution used). All the changes and improvements are marked in red color in the revised manuscript. (Line 381-388)

Point 12: It is not clear to me if the data relative to the extraction of sinapine are relative to a single experiment or not. It seems understood that only proteomic experiments have been repeated. The authors must clarify this point.

Response 12: Thank you very much for your comment. The extractions of sinapine are not a single experiment. The experiments were repeated three times, in triplicate each time.

References

[1] Valentin Reungoat, Morvan Gaudin , Amandine L. Flourat,Emilie Isidore, Louis M.M. Mouterde , Florent Allais , Hélène Ducatel,Irina Ioannou, Optimization of an ethanol/water-based sinapine extraction from mustard bran using Response Surface Methodology, Food and Bioproducts Processing, 2020,122, 322–331.

Reviewer 2 Report

Abstract and Introduction: In both sections, the authors did not describe the objective of their work. Please add the objective in the abstract and introduction.

Why did the authors describe the analysis proteomic as something novelty? I suggest that the proteomics description presented in the introduction section be removed.  

Results:

Figure 1. The authors must describe in a better way the results. For example, they say: “Analysis of sinapine content in rapeseed meal prepared by different extraction techniques. (HE: hot ethanol extraction; HE+UE: Hot ethanol extraction and ultrasonic-assisted extraction; HE+UE+AP: hot ethanol extraction with ultrasonic-assisted extraction combined anti-solvent precipitation). (A) Standard curve for sinapine; (B) sinapine content.” However, this information doesn´t say anything. Fig 1A why did they include a standard curve in the figure? Please remove the standard curve. Figure 1B, 1,5% of the yield of the total sample? Can you explain this result in mg/L?

Figure 2. Before you think about your results in this figure, please let me ask you: what results are you waiting for the viability cell? First, the figure shows that the cells treated (160-640) diminished viability by almost 90% in 24 and 48h. However, you do not show the half-lethal doses or half-lethal concentrations. What means “semi-quantitative analysis”? Statical analysis was deleted from figure 2B?

Again, Figure 2 is not described adequately, please verify the information that you are noticing and describe this figure in the correct manner.        

I´d like to discuss how one compound (sinapine) can regulate the expression of 3554 protein. Please, can you explain this information? 

Figure 3 (figure foot) is moved. 

Figure 6. What do you want to show in the first graph of this section? Please re-organize figure 6A, where you show the western blot bands. 

Please organize the information because the terms “control” and “beta-actin” are very confusing.

The authors noticed the different expressions of proteins, however, they based your analysis on Jack, CDC42, CDC36, and STAT, so why not analyze other proteins different to Jack-stat and CDC42 and 36?   

In general, the authors have interesting results; however, the organization and description are very "fragile". Similar comments for the discussion section, the information is feeble. 

Author Response

Dear Editors and Reviewers,

Thank you for your letter and for the reviewers’ comments concerning our manuscript entitled “Quantitative proteomic analysis reveals the mechanisms of sinapine alleviate macrophage foaming”. Those comments are very helpful for revising and improving our paper. We have studied all issues mentioned in the comments carefully and have corrected. Besides, we have had a check on the whole paper and tried our best to make it better. We have revised the format of the references as required. We hope these meet with approval. Revised portions are marked in red in the manuscript. The main corrections in the paper and the responses to the reviewer’s comments are attached to this revised submission. In addition, all the modifications are highlighted in red in the attachment “main-highlighted”

Below is our careful point-by-point response.

Yours sincerely,

Aiyang Liu

Response to Reviewer 2

Comments

Point 1: Abstract and Introduction: In both sections, the authors did not describe the objective of their work. Please add the objective in the abstract and introduction.

Response 1: Thank you very much for your comment. We have added the objective in the abstract and introduction. The changes and improvements are marked in red color in the revised manuscript. (Line 15-16, Line 51-52)

Point 2: Why did the authors describe the analysis proteomic as something novelty? I suggest that the proteomics description presented in the introduction section be removed.

Response 2: Thank you very much for your comment. We have modified the problem as requested.

Point 3: Figure 1. The authors must describe in a better way the results. For example, they say: “Analysis of sinapine content in rapeseed meal prepared by different extraction techniques. (HE: hot ethanol extraction; HE+UE: Hot ethanol extraction and ultrasonic-assisted extraction; HE+UE+AP: hot ethanol extraction with ultrasonic-assisted extraction combined anti-solvent precipitation). (A) Standard curve for sinapine; (B) sinapine content.” However, this information doesn´t say anything. Fig 1A why did they include a standard curve in the figure? Please remove the standard curve. Figure 1B, 1,5% of the yield of the total sample? Can you explain this result in mg/L?

Response 3: Thank you very much for your comment. We have carefully revised the manuscript according to your comments point by point.

      The sinapine yield (mg/grapeseed meal) is a key parameter for finding the adequate operating conditions of the extraction process [1]. The formula for sinapine yield is as follows:

                                       Yield= (C sinapine *V solvent) /m rapeseed meal

And we multiply the yield result from the above formula by 100%, but to be consistent with the other studied’s units, we have changed the units of yield from (%) to (mg/g). We have added this formula to the manuscript, the modifications are highlighted in red in the pdf file “main-highlighted”. (Line 314-316)

Point 4: Figure 2. Before you think about your results in this figure, please let me ask you: what results are you waiting for the viability cell? First, the figure shows that the cells treated (160-640) diminished viability by almost 90% in 24 and 48h. However, you do not show the half-lethal doses or half-lethal concentrations. What means “semi-quantitative analysis”? Statical analysis was deleted from figure 2B. Again, Figure 2 is not described adequately, please verify the information that you are noticing and describe this figure in the correct manner.  I have redescribed 

Response 4: Thank you very much for your comment. Foam cells were treated with different concentrations of sinapine (0-640 μM) for 24 h and 48 h. After that, the cell viability was detected with a CCK8 kit. The concentration range of sinapine which did not significantly affect the cell viability of macrophage was determined. In the range of 0-80 μM, the viability of the cells was not significantly affected by the treatment when macrophages were treated with sinapine for 24 h, and in the range of 160-640 μM, the viability of the cells was below 100% and significantly different from 80 μM sinapine, and in the range of 0-640 μM, the viability of the cells was below 100% when macrophages were treated with sinapine for 48 h, Therefore, the effect of sinapine on macrophages was investigated further at a concentration of 80 μM after an incubation time of 24 h, and in the CCK8 method [2], no need to show the half-lethal doses or half-lethal concentrations. And to study the effects of sinapine on the lipid content of macrophage. Cells were harvested after culture, fixed with paraformaldehyde, and stained with Oil Red O-Kit. Then Oil Red O staining was measured semi-quantitatively by extracting Oil Red O stain with 100% isopropanol for 10 min gentle rocking and reading absorbance at 492 nm, so semi-quantitatively analysis for the Oil red O staining results, it could be seen that the intracellular lipid content more intuitively. And I have correctly re-described Figure 2, Thank you again for your valuable comments.

Point 5: I´d like to discuss how one compound (sinapine) can regulate the expression of 3554 protein. Please, can you explain this information? 

Figure 3 (figure foot) is moved. 

Response 5: Thank you very much for your comment. Because the cell is a holistic system, when a compound affects the expression of one protein in the cell, it will in turn affect the expression of multiple other proteins. Several studies confirm this, such as a methanolic extract from the Central American plant Lippia origanoides regulates the expression of 3061 protein expression in MDA-MB-231triple-negative breast cancer cells [2]. The compound acts because it affects the expression of several proteins in the cell. And figure 3 (figure foot) is already on a page with figure 3.

Point 6: Figure 6. What do you want to show in the first graph of this section? Please re-organize figure 6A, where you show the western blot bands. 

Please organize the information because the terms “control” and “beta-actin” are very confusing.

Response 6: Thank you very much for your comment. We have re-organized the figure 6A and relevant information, the “control” means the control group in the experiment, “beta-actin” was used as an internal reference protein. We have added it to the manuscript.

Point 7: The authors noticed the different expressions of proteins, however, they based your analysis on Jack, CDC42, CDC36, and STAT, so why not analyze other proteins different from Jack-stat and CDC42 and 36?

Response 7: Thank you very much for your comment. JAK2, CDC42, CD36, and STAT3 were screened by us from several differential proteins, because of their close relationship with macrophages, and they play an important role in regulating cholesterol accumulation in macrophages and can further inhibit the formation of foam cells, so these proteins were selected for further analysis.

Point 8: In general, the authors have interesting results; however, the organization and description are very "fragile". Similar comments for the discussion section, the information is feeble.

Response 8: Thank you very much for your comment. We have studied the organization and description of manuscripts carefully. Besides, we have added some content to the discussion section to make this section fuller and more complete. And we had other teachers and students read the article and corrected some issues. We hope these meet with approval.

References

[1] Valentin Reungoat, Morvan Gaudin , Amandine L. Flourat,Emilie Isidore, Louis M.M. Mouterde, Florent Allais , Hélène Ducatel,Irina Ioannou, Optimization of an ethanol/water-based sinapine extraction from mustard bran using Response Surface Methodology, Food and Bioproducts Processing, 2020,122, 322–331.

[2] Vishak Raman, Uma K. Aryal, Victoria Hedrick, Rodrigo Mohallem Ferreira,

Jorge Luis Fuentes Lorenzo, Elena E. Stashenko, Morris Levy, Maria M. Levy, Ignacio G. Camarillo, Journal of proteome research, 2018,17, 3370-3383.

Reviewer 3 Report

The manuscript entitles "Quantitative proteomic analysis reveals the mechanisms of sinapine alleviate macrophage foaming" investigate a proteomic approach to investigate the effects of sinapine on foam cells.

In my opinion the paper is well structured and written. The results section is well organized and compared with similar researchs in the topic.

The paper shoulb be accepted after minor revisions.

Comments:

- All abbreviations should be described when used for the first time

- The figures quality should be improved so they coulb be more readable, namely figure 4

- The authors should pay attention to significant numbers

- The references are not well formatted

- The number of the reference when cited in the text should be before the point, not after. Please correct.

Author Response

Dear Editors and Reviewers,

Thank you for your letter and for the reviewers’ comments concerning our manuscript entitled “Quantitative proteomic analysis reveals the mechanisms of sinapine alleviate macrophage foaming”. Those comments are very helpful for revising and improving our paper. We have studied all issues mentioned in the comments carefully and have corrected. Besides, we have had a check on the whole paper and tried our best to make it better. We have revised the format of the references as required. We hope these meet with approval. Revised portions are marked in red in the manuscript. The main corrections in the paper and the responses to the reviewer’s comments are attached to this revised submission. In addition, all the modifications are highlighted in red in the attachment “main-highlighted”

Below is our careful point-by-point response.

Yours sincerely,

Aiyang Liu

Response to Reviewer 3

Comments

Point 1: All abbreviations should be described when used for the first time

Response 1: Thank you very much for your comments. We have modified the problem as requested.

Point 2: The figures quality should be improved so they coulb be more readable, namely figure 4

Response 2: Thank you very much for your comments. We have improved the figures quality as requested.

Point 3: The authors should pay attention to significant numbers

Response 3: Thank you very much for your comments. We have modified the problem as requested.

Point 4: The references are not well formatted

Response 4: Thank you very much for your comments. We have revised the format of the references as required.

Point 5: The number of the reference when cited in the text should be before the point, not after. Please correct.

Response 5: Thank you very much for your comments. We have modified the problem as requested.

Round 2

Reviewer 1 Report

I fully understand that English is not the mother tongue of the authors but the reader has the right to read an understandable text. I, as referee, have the duty to point out the language inconsistencies still present in this text.

For example lines 49-59 are totally gibberish. Is it possible that the authors do not notice this? 

line 17: should read ......."than in traditional......"

line 18: should become "....  on foam cells. We found ...."

lines 64-65: this sentence must be improved.

and many other inconsistencies throughout the text.

Author Response

Dear Editors and Reviewers,

Thank you for your letter and for the reviewers’ comments concerning our manuscript entitled “Quantitative proteomic analysis reveals the mechanisms of sinapine alleviate macrophage foaming”. Those comments are very helpful for revising and improving our paper. We have studied all issues mentioned in the comments carefully and have corrected. Besides, we have checked on the whole paper and tried our best to make it better. We have revised the format of the references as required. We hope these meet with approval. Revised portions are marked in red in the manuscript. The main corrections in the paper and the responses to the reviewer’s comments are attached to this revised submission. In addition, all the modifications are highlighted in red in the attachment “molecules-2193579 - highlight”

Below is our careful point-by-point response.

Yours sincerely,

Aiyang Liu

Response to Reviewer 1 Comments

Point 1: For example lines 49-59 are totally gibberish. Is it possible that the authors do not notice this?

Response 1: Thank you very much for your comments. We have modified the problem as requested. All the changes and improvements are marked in red color in the revised manuscript. (Line 46-53)

Point 2: line 18: should become "....  on foam cells. We found ...."

Response 2: Thank you very much for your comment. We have now modified this into: "....  on foam cells. We found .....". The modifications are highlighted in red in the pdf file “main-highlighted”.(Line 17-18)

Point 3: lines 64-65: this sentence must be improved.

Response 3: Thank you very much for your comment. We have now modified this into: “higher than that of”. The modifications are highlighted in red in the pdf file “main-highlighted”.(Line 73-74)

Point 4: and many other inconsistencies throughout the text.

Response 4: Thank you very much for your comment. We have revised the paper based on your instruction, and improvements are marked in red color in the revised manuscript.
